# Poisson-Randomized Gamma Dynamical Systems

**Aaron Schein**
Data Science Institute
Columbia University

**Scott W. Linderman**
Department of Statistics
Stanford University

**Mingyuan Zhou**
McCombs School of Business
University of Texas at Austin

**David M. Blei**
Department of Statistics
Columbia University

**Hanna Wallach**
Microsoft Research
New York, NY

## Abstract

This paper presents the Poisson-randomized gamma dynamical system (PRGDS), a model for sequentially observed count tensors that encodes a strong inductive bias toward sparsity and burstiness. The PRGDS is based on a new motif in Bayesian latent variable modeling, an alternating chain of discrete Poisson and continuous gamma latent states that is analytically convenient and computationally tractable. This motif yields closed-form complete conditionals for all variables by way of the Bessel distribution and a novel discrete distribution that we call the shifted confluent hypergeometric distribution. We draw connections to closely related models and compare the PRGDS to these models in studies of real-world count data sets of text, international events, and neural spike trains. We find that a sparse variant of the PRGDS, which allows the continuous gamma latent states to take values of exactly zero, often obtains better predictive performance than other models and is uniquely capable of inferring latent structures that are highly localized in time.

## 1 Introduction

Political scientists routinely analyze event counts of the number of times country $i$ took action $a$ toward country $j$ during time step $t$ [1]. Such data can be represented as a sequence of count tensors $\boldsymbol{Y}^{(1)}, \ldots, \boldsymbol{Y}^{(T)}$ each of which contains the $V \times V \times A$ event counts for that time step for every combination of $V$ sender countries, $V$ receivers, and $A$ action types. International event data sets exhibit "complex dependence structures" [2] like coalitions of countries and bursty temporal dynamics. These dependence structures violate the independence assumptions of the regression-based methods that political scientists have traditionally used to test theories of international relations [3–5]. Political scientists have therefore advocated for using latent variable models to infer unobserved structures as a way of controlling for them [6]. This approach motivates interpretable yet expressive models that are capable of capturing a variety of complex dependence structures. Recent work has applied tensor factorization methods to international event data sets [7–11] to infer coalition structures among countries and topic structures among actions; however, these methods assume that the sequentially observed count tensors are exchangeable, thereby failing to capture the bursty temporal dynamics inherent to such data sets.

Sequentially observed count tensors present unique statistical challenges because they tend to be bursty [12], high-dimensional, and sparse [13, 14]. There are few models that are tailored to the challenging properties of both time series and count tensors. In recent years, Poisson factorization has emerged as a framework for modeling count matrices [15–20] and tensors [13, 21, 9]. Although factorization methods generally scale with the size of the matrix or tensor, many Poisson factorization models yield inference algorithms that scale linearly with the number of non-zero entries. This property allows researchers to efficiently infer latent structures from massive tensors, provided these tensors are sparse; however, this property is unique to a subset of Poisson factorization models that only posit

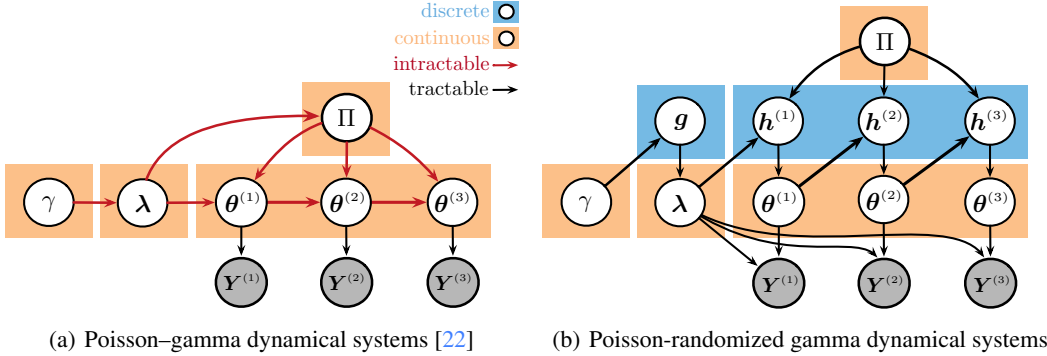

(a) Poisson–gamma dynamical systems [22]    (b) Poisson-randomized gamma dynamical systems

Figure 1: *Left*: The PGDS imposes dependencies directly between the gamma latent states, preventing closed-form complete conditionals. *Right*: The PRGDS (this paper) breaks these dependencies with discrete Poisson latent states—doing so yields closed-form conditionals for all variables without data augmentation.

non-negative prior distributions, which are difficult to chain in state-space models for time series. Hierarchical compositions of non-negative priors—notably, gamma and Dirichlet distributions—typically introduce non-conjugate dependencies that require innovative approaches to posterior inference.

This paper fills a gap in the literature between Poisson factorization models that are tractable—i.e., yielding closed-form complete conditionals that make inference algorithms easy to derive—and those that are expressive—i.e., capable of capturing a variety of complex dependence structures. To do so, we introduce an alternating chain of discrete Poisson and continuous gamma latent states, a new modeling motif that is analytically convenient and computationally tractable. We rely on this motif to construct the Poisson-randomized gamma dynamical system (PRGDS), a model for sequentially observed count tensors that is tractable, expressive, and efficient. The PRGDS is closely related to the Poisson–gamma dynamical system (PGDS) [22], a recently introduced model for dynamic count matrices, that is based on non-conjugate chains of gamma states. These chains are intractable; thus, posterior inference in the PGDS relies on sophisticated data augmentation schemes that are cumbersome to derive and impose unnatural restrictions on the priors over other variables. In contrast, the PRGDS introduces intermediate Poisson states that break the intractable dependencies between the gamma states (see Fig. 1). Although this motif is only semi-conjugate, it is tractable, yielding closed-form complete conditionals for the Poisson states by way of the little-known Bessel distribution [23] and a novel discrete distribution that we derive and call the *shifted confluent hypergeometric (SCH) distribution*.

We study the inductive bias of the PRGDS by comparing its smoothing and forecasting performance to that of the PGDS and two other baselines on a range of real-world count data sets of text, international events, and neural spike data. For smoothing, we find that the PRGDS performs better than or similarly to the PGDS; for forecasting, we find the converse relationship. Both models outperform the other baselines. Using a specific hyperparameter setting, the PRGDS permits the continuous gamma latent states to take values of exactly zero, thereby encoding a unique inductive bias tailored to sparsity and burstiness. We find that this sparse variant always obtains better smoothing and forecasting performance than the non-sparse variant. We also find that this sparse variant yields a qualitatively broader range of latent structures—specifically, bursty latent structures that are highly localized in time.

## 2  Poisson-randomized gamma dynamical systems (PRGDS)

**Notation.** Consider a data set of sequentially observed count tensors $\boldsymbol{Y}^{(1)}, \ldots, \boldsymbol{Y}^{(T)}$, each of which has $M$ modes. An entry $y_{\mathbf{i}}^{(t)} \in \{0, 1, 2, \ldots\}$ in the $t^{\text{th}}$ tensor is subscripted by a multi-index $\mathbf{i} \equiv (\mathrm{i}_1, \ldots, \mathrm{i}_M)$ that indexes into the $M$ modes of the tensor. As an example, the event count of the number of times country $i$ took action $a$ toward country $j$ during time step $t$ can be written as $y_{\mathbf{i}}^{(t)}$ where the multi-index corresponds to the sender, receiver, and action type—i.e., $\mathbf{i} = (i, j, a)$.

**Generative process.** The PRGDS is a form of canonical polyadic decomposition [24] that assumes

$$y_{\mathbf{i}}^{(t)} \sim \text{Pois}\Big(\rho^{(t)} \sum_{k=1}^{K} \lambda_k \, \theta_k^{(t)} \prod_{m=1}^{M} \phi_{k\mathrm{i}_m}^{(m)}\Big), \tag{1}$$

where $\theta_k^{(t)}$ represents the activation of the $k^{\text{th}}$ component at time step $t$. Each component represents a dependence structure in the data set by way of a factor vector $\boldsymbol{\phi}_k^{(m)}$ for each mode $m$. For international events, the first factor vector $\boldsymbol{\phi}_k^{(1)} = (\phi_{k1}^{(1)}, \ldots, \phi_{kV}^{(1)})$ represents the rate at which each of the $V$ countries acts as a sender in the $k^{\text{th}}$ component while the second factor vector $\boldsymbol{\phi}_k^{(2)}$ represents the rate at which each country acts as a receiver. The weights $\lambda_k$ and $\rho^{(t)}$ represent the scales of component $k$ and time step $t$. The PRGDS is stationary if $\rho^{(t)} = \rho$. We posit the following conjugate priors:

$$\rho^{(t)} \sim \mathrm{Gam}\left(a_0, b_0\right) \quad \text{and} \quad \boldsymbol{\phi}_k^{(m)} \sim \mathrm{Dir}(a_0, \ldots, a_0). \tag{2}$$

The PRGDS is characterized by an alternating chain of discrete and continuous latent states. The continuous states $\theta_k^{(1)}, \ldots, \theta_k^{(T)}$ evolve via the intermediate discrete states $h_k^{(1)}, \ldots, h_k^{(T)}$ as follows:

$$\theta_k^{(t)} \sim \mathrm{Gam}\left(\epsilon_0^{(\theta)} + h_k^{(t)}, \tau\right) \quad \text{and} \quad h_k^{(t)} \sim \mathrm{Pois}\left(\tau \sum_{k_2=1}^{K} \pi_{kk_2} \theta_{k_2}^{(t-1)}\right), \tag{3}$$

where we define $\theta_k^{(0)} = \lambda_k$ to be the per-component weight from Eq. (1). In other words, the PRGDS assumes that $\theta_k^{(t)}$ is conditionally gamma distributed with rate $\tau$ and shape equal to $h_k^{(t)}$ plus hyperparameter $\epsilon_0^{(\theta)} \geq 0$. We adopt the convention that a gamma random variable will be zero, almost surely, if its shape is zero. Therefore, setting $\epsilon_0^{(\theta)} = 0$ defines a sparse variant of the PRGDS, where the gamma latent state $\theta_k^{(t)}$ takes the value of exactly zero provided $h_k^{(t)} = 0$—i.e., $\theta_k^{(t)} \overset{\text{a.s.}}{=} 0$ if $h_k^{(t)} = 0$.

The *transition weight* $\pi_{kk_2}$ in Eq. (3) represents how strongly component $k_2$ excites component $k$ at the next time step. We view these weights collectively as a $K \times K$ transition matrix $\Pi$ and impose Dirichlet priors over the columns of this matrix. We also place a gamma prior over concentration parameter $\tau$. This prior is conjugate to the gamma and Poisson distributions in which it appears:

$$\tau \sim \mathrm{Gam}\left(\alpha_0, \alpha_0\right) \quad \text{and} \quad \boldsymbol{\pi}_k \sim \mathrm{Dir}\left(a_0, \ldots, a_0\right) \text{ such that } \sum_{k_1=1}^{K} \pi_{k_1 k} = 1. \tag{4}$$

For the per-component weights $\lambda_1, \ldots, \lambda_K$, we use a hierarchical prior with a similar flavor to Eq. (3):

$$\lambda_k \sim \mathrm{Gam}\left(\frac{\epsilon_0^{(\lambda)}}{K} + g_k, \beta\right) \quad \text{and} \quad g_k \sim \mathrm{Pois}\left(\frac{\gamma}{K}\right), \tag{5}$$

where $\epsilon_0^{(\lambda)}$ is analogous to $\epsilon_0^{(\theta)}$. Finally, we use the following gamma priors, which are both conjugate:

$$\gamma \sim \mathrm{Gam}\left(a_0, b_0\right) \quad \text{and} \quad \beta \sim \mathrm{Gam}\left(\alpha_0, \alpha_0\right). \tag{6}$$

The PRGDS has five fixed hyperparameters: $\epsilon_0^{(\theta)}$, $\epsilon_0^{(\lambda)}$, $\alpha_0$, $a_0$, and $b_0$. For the empirical studies in § 5, we set $a_0 = b_0 = 0.01$ to define weakly informative gamma and Dirichlet priors and set $\alpha_0 = 10$ to define a gamma prior that promotes values close to 1; we consider $\epsilon_0^{(\theta)} \in \{0, 1\}$ and set $\epsilon_0^{(\lambda)} = 1$.

**Properties.** In Eq. (5), both $\epsilon_0^{(\lambda)}$ and $\gamma$ are divided by the number of components $K$. This means that as the number of components grows $K \to \infty$, the expected sum of the weights remains finite and fixed:

$$\sum_{k=1}^{\infty} \mathbb{E}\left[\lambda_k\right] = \sum_{k=1}^{\infty} \left(\frac{\epsilon_0^{(\lambda)}}{K} + \mathbb{E}\left[g_k\right]\right)\beta^{-1} = \sum_{k=1}^{\infty} \left(\frac{\epsilon_0^{(\lambda)}}{K} + \frac{\gamma}{K}\right)\beta^{-1} = \left(\epsilon_0^{(\lambda)} + \gamma\right)\beta^{-1}. \tag{7}$$

This prior encodes an inductive bias toward small values of $\lambda_k$ and may be interpreted as the finite truncation of a novel Bayesian nonparametric process. A small value of $\lambda_k$ shrinks the Poisson rates of both $y_{\mathbf{i}}^{(t)}$ and the first discrete latent state $h_k^{(0)}$. As a result, this prior encourages the PRGDS to only infer components that are both predictive of the data and useful for capturing the temporal dynamics.

The marginal expectation of $\boldsymbol{\theta}^{(t)} = (\theta_1^{(t)}, \ldots, \theta_K^{(t)})$ takes the form of a linear dynamical system:

$$\mathbb{E}\left[\boldsymbol{\theta}^{(t)} \mid \boldsymbol{\theta}^{(t-1)}\right] = \mathbb{E}\left[\mathbb{E}\left[\boldsymbol{\theta}^{(t)} \mid \boldsymbol{h}^{(t-1)}\right]\right] = \epsilon_0^{(\theta)} \tau^{-1} + \Pi \boldsymbol{\theta}^{(t-1)}. \tag{8}$$

This is because $\mathbb{E}\left[\theta_k^{(t)}\right] = \left(\epsilon_0^{(\theta)} + \mathbb{E}\left[h_k^{(t)}\right]\right)\tau^{-1} = \left(\epsilon_0^{(\theta)} + \tau \sum_{k_2=1}^{K} \pi_{kk_2} \theta_{k_2}^{(t)}\right)\tau^{-1}$ by iterated expectation. Concentration parameter $\tau$ appears in both the Poisson and gamma distributions in Eq. (3). It contributes to the variance of the PRGDS, while simultaneously canceling out of the expectation in Eq. (8), except for its role in the additive term $\epsilon_0^{(\theta)} \tau^{-1}$, which itself disappears when $\epsilon_0^{(\theta)} = 0$.

Finally, we can analytically marginalize out all of the discrete Poisson latent states to obtain a purely continuous dynamical system. When $\epsilon_0^{(\theta)} > 0$, this dynamical system can be written as follows:

$$\theta_k^{(t)} \sim \mathrm{RG1}\left(\epsilon_0^{(\theta)}, \tau \sum_{k_2=1}^{K} \pi_{kk_2} \theta_{k_2}^{(t-1)}, \tau\right), \tag{9}$$

where RG1 denotes the randomized gamma distribution of the first type [23, 25]. When $\epsilon_0^{(\theta)} = 0$, the dynamical system can be written in terms of a limiting form of the RG1. We describe the RG1 in Fig. 2.

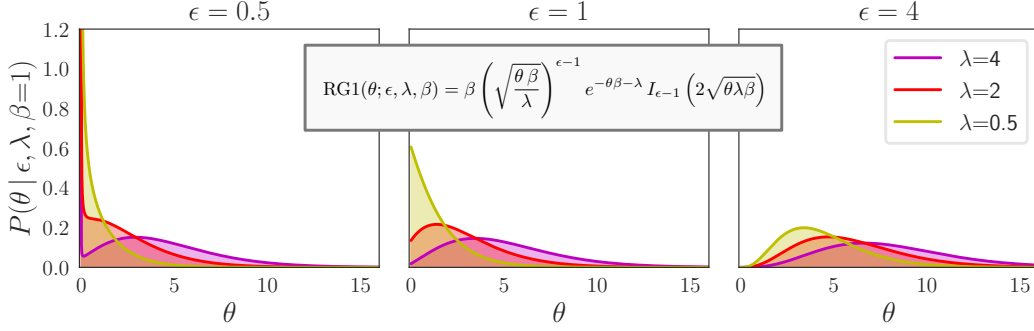

Figure 2: The randomized gamma distribution of the first type (RG1) [23, 25] has support $\theta > 0$ and is defined by three parameters: $\epsilon, \lambda, \beta > 0$. Its PDF is displayed in the figure; $I_{\epsilon-1}(\cdot)$ is the modified Bessel function of the first kind [26]. When $\epsilon < 1$ (*left*), the RG1 resembles a soft "spike-and-slab" distribution; when $\epsilon \geq 1$ (*middle and right*), it resembles a more-dispersed form of the gamma distribution. The Poisson-randomized gamma distribution [27], which includes zeros in its support (i.e., $\theta \geq 0$), is a limiting case of the RG1 that occurs when $\epsilon \to 0$.

## 3 Related work

The PRGDS is closely related to the Poisson–gamma dynamical system (PGDS) [22]. In the PGDS,

$$\theta_k^{(t)} \sim \text{Gam}\Big(\tau \sum_{k_2=1}^{K} \pi_{kk_2} \theta_{k_2}^{(t-1)}, \tau\Big) \quad \text{such that} \quad \mathbb{E}\big[\boldsymbol{\theta}^{(t)} \,|\, \boldsymbol{\theta}^{(t-1)}\big] = \Pi \boldsymbol{\theta}^{(t-1)}. \tag{10}$$

The PGDS imposes non-conjugate dependencies directly between the gamma latent states. The complete conditional $P(\theta_k^{(t)}|-)$ is not available in closed form, and posterior inference relies on a sophisticated data augmentation scheme. The PRGDS instead introduces intermediate Poisson states that break the intractable dependencies between the gamma states; we visualize this in Fig. 1. Although the Poisson distribution is not a conjugate prior for the gamma rate, this motif is still tractable, yielding the complete conditional $P(h_k^{(t)}|-)$ in closed form, as we explain in § 4. The PGDS is limited by the data augmentation scheme that it relies on for posterior inference—specifically, this augmentation scheme does not allow $\lambda_k$ to appear in the Poisson rate of $y_{\mathbf{i}}^{(t)}$ in Eq. (1). To encourage parsimony, the PGDS instead draws $\lambda_k \sim \text{Gam}(\frac{\gamma}{K}, \beta)$ and then uses these per-component weights to shrink the transition matrix $\Pi$. This approach introduces additional intractable dependencies that require a different data augmentation scheme for posterior inference. Finally, the data augmentation schemes additionally require that each factor vector $\boldsymbol{\phi}_k^{(m)}$ and each column $\boldsymbol{\pi}_k$ of the transition matrix are Dirichlet distributed. We note that although we also use Dirichlet distributions in this paper, this is a choice rather than a requirement imposed by the PRGDS.

The PGDS and its "deep" variants [28, 29] generalize gamma process dynamic Poisson factor analysis (GP-DPFA) [30], which assumes a simple random walk $\theta_k^{(t)} \sim \text{Gam}\big(\theta_k^{(t-1)}, c^{(t)}\big)$; the model of Yang and Koeppl is also closely related [31]. These models belong to a line of work exploring the "augment-and-conquer" data augmentation scheme [32] for posterior inference in hierarchies of gamma variables chained via their shapes and linked to Poisson observations. Beyond models for time series, this motif can be used to build belief networks [33]. An alternative approach is to chain gamma variables via their rates—e.g., $\theta^{(t)} \sim \text{Gam}\big(a, \theta^{(t-1)}\big)$. This motif is conjugate and tractable, and has been applied to models for time series [34–36] and deep belief networks [37]. However, unlike the shape, the rate contributes to the variance of the gamma quadratically. Rate chains can therefore be highly volatile.

More broadly, gamma shape and rate chains are examples of non-negative chains. Such chains are especially well motivated in the context of Poisson factorization, which is particularly efficient when only non-negative prior distributions are used. In general, Poisson factorization assumes that each observed count $y_{\mathbf{i}}$ is drawn from a Poisson distribution with a latent rate $\mu_{\mathbf{i}}$ that is some function of the model parameters—i.e., $y_{\mathbf{i}} \sim \text{Pois}(\mu_{\mathbf{i}})$. When the rate is linear—i.e., $\mu_{\mathbf{i}} = \sum_{k=1}^{K} \mu_{\mathbf{i}k}$—Poisson factorization is allocative [38] and admits a latent source representation [16, 18], where $y_{\mathbf{i}} \triangleq \sum_{k=1}^{K} y_{\mathbf{i}k}$ is defined to be the sum of $K$ latent sources $y_{\mathbf{i}1}, \dots, y_{\mathbf{i}K}$ and $y_{\mathbf{i}k} \sim \text{Pois}(\mu_{\mathbf{i}k})$. Conditioning on the latent sources often induces conditional independencies that, in turn, facilitate closed-form, efficient, and parallelizable posterior inference. The first step in either MCMC or variational inference

is therefore to update each latent source from its complete conditional, which is multinomial [39]:

$$\left((y_{\mathbf{i}1},\ldots,y_{\mathbf{i}K})\,|\,-\,\right) \sim \mathrm{Multinom}\left(y_{\mathbf{i}},\,(\mu_{\mathbf{i}1},\ldots,\mu_{\mathbf{i}K})\right), \qquad (11)$$

where the normalization of the non-negative rates $\mu_{\mathbf{i}1},\ldots,\mu_{\mathbf{i}K}$ into a probability vector is left implicit. When the observed count is zero—i.e., $y_{\mathbf{i}}=0$—the sources are also zero—i.e., $y_{\mathbf{i}k} \overset{\mathrm{a.s.}}{=} 0$— and no computation is required to update them. As a result, any Poisson factorization model that admits a latent source representation scales linearly with only the non-zero entries. This property is indispensable when modeling count tensors which typically contain exponentially more zeros than non-zeros [40]. We emphasize that although the PRGDS and PGDS are substantively different models, they are both instances of allocative Poisson factorization, so the time complexity of posterior inference for both models is the same and equal to $\mathcal{O}\left(SK\right)$ where $S$ is the number of non-zero entries.

Because a latent source representation is only available when the rate $\mu_{\mathbf{i}}$ is a linear function of the model parameters and, by definition of the Poisson distribution, the rate must be non-negative, efficient Poisson factorization is only possible with non-negative priors. Modeling time series and other complex dependence structures via efficient Poisson factorization therefore requires developing novel motifs that exclude the Gaussian priors that researchers have traditionally relied on for analytic convenience and tractability. For example, the Poisson linear dynamical system [41–43] links the widely used Gaussian linear dynamical system [44, 45] to Poisson observations via an exponential link function—i.e., $\mu_{\mathbf{i}} = \exp\left(\sum_k \cdots\right)$. This approach, which is based on the generalized linear model [46], relies on a non-linear link function and therefore does not admit a latent source representation. Another approach is to use log-normal priors, as in dynamic Poisson factorization [47]; however, the log-normal is not conjugate to the Poisson distribution and does not yield closed-form conditionals.

There is also a long tradition of autoregressive models for time series of counts, including variational autoregressive models [48] and models that are based on the Hawkes process [49–52]. This approach avoids the challenge of constructing tractable state-space models from non-negative priors by modeling temporal correlations directly between the observed counts. However, for high-dimensional data, such as sequentially observed count tensors, an autoregressive approach is often impractical.

## 4 Posterior inference

Iteratively re-sampling each latent variable in the PRGDS from its complete conditional constitutes a Gibbs sampling algorithm. The complete conditionals for all variables are immediately available in closed form without data augmentation. We provide conditionals for the variables with non-standard priors below; the remaining conditionals are in the supplementary material. The PRGDS is based on a new motif in Bayesian latent variable modeling. We introduce the motif in its general form, derive its conditionals, and then use these to obtain the closed-form complete conditionals for the PRGDS.

### 4.1 Poisson–gamma–Poisson chains

Consider the following model of count $m$ involving variables $\theta$ and $h$ and fixed $c_1, c_2, c_3, \epsilon_0^{(\theta)} > 0$:

$$m \sim \mathrm{Pois}\left(\theta c_3\right),\;\; \theta \sim \mathrm{Gam}\left(\epsilon_0^{(\theta)}+h, c_2\right),\;\; \text{and}\;\; h \sim \mathrm{Pois}\left(c_1\right). \qquad (12)$$

This model is semi-conjugate. The gamma prior over $\theta$ is conjugate to the Poisson and its posterior is

$$\left(\theta\,|\,-\,\right) \sim \mathrm{Gam}\left(\epsilon_0^{(\theta)}+h+m,\, c_2+c_3\right). \qquad (13)$$

The Poisson prior over $h$ is not conjugate to the gamma; however, despite this, the posterior of $h$ is still available in closed form by way of the Bessel distribution [23], which we define in Fig. 3(a):

$$\left(h\,|\,-\,\right) \sim \mathrm{Bes}\left(\epsilon_0^{(\theta)}-1,\, 2\sqrt{\theta\,c_2\,c_1}\right). \qquad (14)$$

The Bessel distribution can be sampled efficiently [53]; our Cython implementation is available online.[1] Provided that $\epsilon_0^{(\theta)} > 0$, sampling $\theta$ and $h$ iteratively from Eqs. (13) and (14) constitutes a valid Markov chain for posterior inference. When $\epsilon_0^{(\theta)}=0$, though, $\theta \overset{\mathrm{a.s.}}{=} 0$ if $h=0$, and vice versa. As a result, this Markov chain has an absorbing condition at $h=0$ and violates detailed balance. In this case, we must therefore sample $h$ with $\theta$ marginalized out. Toward that end, we prove Theorem 1.

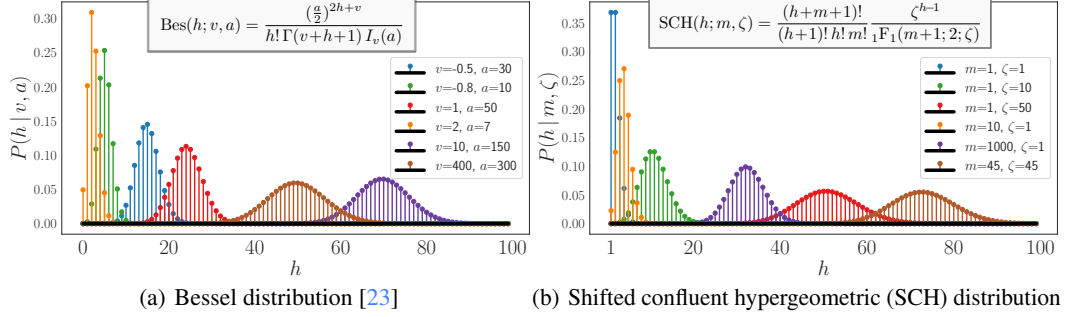

(a) Bessel distribution [23]     (b) Shifted confluent hypergeometric (SCH) distribution

Figure 3: Two discrete distributions that arise as posteriors in Poisson–gamma–Poisson chains.

**Theorem 1:** *The incomplete conditional* $P(h \mid \epsilon_0^{(\theta)} = 0, -\backslash\theta) \triangleq \int P(h, \theta \mid \epsilon_0^{(\theta)} = 0, -) \, \mathrm{d}\theta$ *is*

$$(h \mid -\backslash\theta) \sim \begin{cases} \mathrm{Pois}\left(\frac{c_1\,c_2}{c_3+c_2}\right) & \text{if } m = 0 \\ \mathrm{SCH}\left(m, \frac{c_1\,c_2}{c_3+c_2}\right) & \text{otherwise,} \end{cases} \tag{15}$$

*where SCH denotes the shifted confluent hypergeometric distribution. We describe the SCH in Fig. 3(b) and provide further information in the supplementary material, including the derivation of its PMF, PGF, and mode, along with details of how we sample from it and the proof for Theorem 1.*

### 4.2 Closed-form complete conditionals for the PRGDS

The PRGDS admits a latent source representation, so the first step of posterior inference is therefore

$$\left((y_{\mathbf{i}k}^{(t)})_{k=1}^K \mid -\right) \sim \mathrm{Multinom}\left(y_{\mathbf{i}}^{(t)}, \left(\lambda_k\,\theta_k^{(t)}\prod_{m=1}^M \phi_{k\mathbf{i}_m}^{(m)}\right)_{k=1}^K\right). \tag{16}$$

We may similarly represent $h_k^{(t)}$ under its latent source representation—i.e., $h_k^{(t)} \equiv h_{k\cdot}^{(t)} = \sum_{k_2=1}^K h_{kk_2}^{(t)}$, where $h_{kk_2}^{(t)} \sim \mathrm{Pois}\left(\tau\,\pi_{kk_2}\theta_{k_2}^{(t-1)}\right)$. When notationally convenient, we use dot-notation ("$\cdot$") to denote summing over a mode. In this case, $h_{k\cdot}^{(t)}$ denotes the sum of the $k^{\text{th}}$ row of the $K \times K$ matrix of latent counts $h_{kk_2}^{(t)}$. The complete conditional of the $k^{\text{th}}$ row of counts, when conditioned on their sum $h_{k\cdot}^{(t)}$, is

$$\left((h_{kk_2}^{(t)})_{k_2=1}^K \mid -\right) \sim \mathrm{Multinom}\left(h_{k\cdot}^{(t)}, (\pi_{kk_2}\theta_{k_2}^{(t-1)})_{k_2=1}^K\right). \tag{17}$$

To derive the conditional for $\theta_k^{(t)}$ we aggregate the Poisson variables that depend on it. By Poisson additivity, the column sum $h_{\cdot k}^{(t+1)} = \sum_{k_1=1}^K h_{k_1 k}^{(t+1)}$ is distributed as $h_{\cdot k}^{(t+1)} \sim \mathrm{Pois}\left(\theta_k^{(t)}\,\tau\,\pi_{\cdot k}\right)$ and similarly $y_{\cdot k}^{(t)}$ is distributed as $y_{\cdot k}^{(t)} \sim \mathrm{Pois}\left(\theta_k^{(t)}\rho^{(t)}\lambda_k\prod_{m=1}^M \phi_{\cdot k\cdot}^{(m)}\right)$. The count $m_k^{(t)} \triangleq h_{\cdot k}^{(t+1)} + y_{\cdot k}^{(t)}$ isolates all dependence on $\theta_k^{(t)}$ and is also Poisson distributed. By gamma–Poisson conjugacy, the conditional of $\theta_k^{(t)}$ is

$$\left(\theta_k^{(t)} \mid -\right) \sim \mathrm{Gam}\left(\epsilon_0^{(\theta)} + h_{k\cdot}^{(t)} + m_k^{(t)},\ \tau + \tau\,\pi_{\cdot k} + \rho^{(t)}\lambda_k\prod_{m=1}^M \phi_{k\cdot}^{(m)}\right). \tag{18}$$

When $\epsilon_0^{(\theta)} > 0$, we apply the identity in Eq. (14) and sample $h_{k\cdot}^{(t)}$ from its complete conditional:

$$\left(h_{k\cdot}^{(t)} \mid -\right) \sim \mathrm{Bessel}\left(\epsilon_0^{(\theta)} - 1,\ 2\sqrt{\theta_k^{(t)}\,\tau^2\sum_{k_2=1}^K \pi_{kk_2}\theta_{k_2}^{(t-1)}}\right). \tag{19}$$

When $\epsilon_0^{(\theta)} = 0$, we instead apply Theorem 1 to sample $h_{k\cdot}^{(t)}$, where $m_k^{(t)}$ is analogous to $m$ in Eq. (15):

$$\left(h_{k\cdot}^{(t)} \mid -\backslash\theta_k^{(t)}\right) \sim \begin{cases} \mathrm{Pois}(\zeta_k^{(t)}) & \text{if } m_k^{(t)} = 0 \\ \mathrm{SCH}(m_k^{(t)}, \zeta_k^{(t)}) & \text{otherwise} \end{cases} \quad \text{where } \zeta_k^{(t)} \triangleq \frac{\tau^2\sum_{k_2=1}^K \pi_{kk_2}\theta_{k_2}^{(t-1)}}{\tau + \tau\,\pi_{\cdot k} + \rho^{(t)}\lambda_k\prod_{m=1}^M \phi_{k\cdot}^{(m)}}. \tag{20}$$

The complete conditionals for $\lambda_k$ and $g_k$ follow from applying the same Poisson–gamma–Poisson identities, while the complete conditionals for $\gamma$, $\beta$, $\phi_k^{(m)}$, $\pi_k$, and $\tau$ all follow from conjugacy.

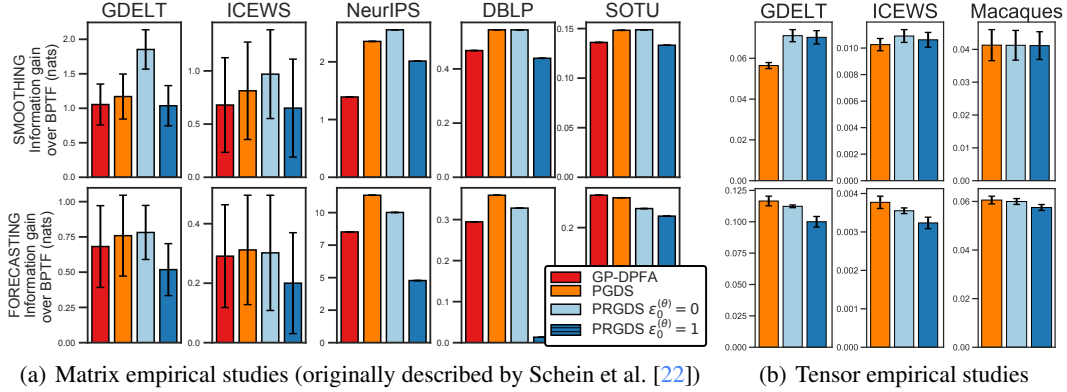

(a) Matrix empirical studies (originally described by Schein et al. [22])   (b) Tensor empirical studies

Figure 4: The smoothing performance (top row) or forecasting performance (bottom row) of each model is quantified by its information gain over a non-dynamic baseline (BPTF [9]), where higher values are better.

## 5  Empirical studies

As explained in the previous section, the Poisson–gamma–Poisson motif of the PRGDS (see § 4.1) yields a more tractable (see Fig. 1) and flexible (see § 3) model than previous models. This motif also encodes a unique inductive bias tailored to sparsity and burstiness that we test by comparing the PRGDS to the PGDS (described in § 3). As we can see by comparing Eqs. (9) and (10), comparing these models isolates the impact of the Poisson–gamma–Poisson motif. Because the PGDS was previously introduced to model a $T \times V$ matrix $Y$ of sequentially observed $V$-dimensional count vectors $\boldsymbol{y}^{(1)}, \ldots, \boldsymbol{y}^{(T)}$, we generalize the PGDS to $M$-mode tensors and provide derivations of its complete conditionals in the supplementary material. Our Cython implementation of this generalized PGDS (and the PRGDS) is available online. We also compare the variant of the PRGDS with $\epsilon_0^{(\theta)} = 1$ to the variant with $\epsilon_0^{(\theta)} = 0$, which allows the continuous gamma latent states to take values of exactly zero.

**Setup.**  Our empirical studies all have the following setup. For each data set $\boldsymbol{Y}^{(1)}, \ldots, \boldsymbol{Y}^{(T)}$, the counts $\boldsymbol{Y}^{(t)}$ in randomly selected time steps are held out. Additionally, the counts in the last two time steps are always held out. Each model is fit to the data set using independent MCMC chains that impute the heldout counts and, ultimately, return a set of posterior samples of the latent variables. We distinguish the task of predicting the counts in intermediate time steps, known as smoothing, from the task of predicting the counts in the last two time steps, known as forecasting. To quantify the performance of each model, we use the $S$ posterior samples returned by the independent chains to approximate the information rate [54] of the heldout counts—i.e.,

$R(\Delta) = -\frac{1}{|\Delta|} \sum_{(t,\mathbf{i}) \in \Delta} \log \left[ \frac{1}{S} \sum_{s=1}^{S} \mathrm{Pois}\left(y_{\mathbf{i}}^{(t)}; \mu_{\mathbf{i},s}^{(t)}\right) \right]$, where $\Delta$ is the set of multi-indices of the

heldout counts and $\mu_{\mathbf{i},s}^{(t)}$ is the expectation of heldout count $y_{\mathbf{i}}^{(t)}$ (defined in Eq. (1)) computed from the $s^{\text{th}}$ posterior sample. The information rate quantifies the average number of nats needed to compress each heldout count; it is equivalent to log perplexity [55] and to the negative of log pointwise predictive density (LPPD) [56]. In each study, we also fit Bayesian Poisson tensor factorization (BPTF) [9], a non-dynamic baseline that assumes that the count tensors at different time steps are i.i.d.—i.e., $y_{\mathbf{i}}^{(t)} \sim \mathrm{Pois}(\mu_{\mathbf{i}})$. For each model, we then report the information gain over BPTF, where higher values are better, which we compute by subtracting the information rate of the model from that of BPTF.

**Matrices.**  We first replicated the empirical studies of Schein et al. [22]. These studies followed the setup described above and compared the PGDS to GP-DPFA [30], a simple dynamic baseline (described in § 3). The matrices in these studies were based on three text data sets—NeurIPS papers [57], DBLP abstracts [58], and State of the Union (SOTU) speeches [59]—where $y_v^{(t)}$ is the number of times word $v$ occurs in time step $t$, and two international event data sets—GDELT [60] and ICEWS [61]—where $y_v^{(t)}$ is the number of times sender–receiver pair $v$ interacted during time step $t$. We used the matrices and heldout time steps, along with the posterior samples for both PGDS and GP-DPFA, originally obtained by Schein et al. [22]. We then fit the PRGDS using the MCMC settings that they describe. In this matrix setting, BPTF reduces to $y_v^{(t)} \sim \mathrm{Pois}(\mu_v)$, where $v$ indexes a single mode, and $\mu_v$ cannot be meaningfully factorized. We therefore posited a conjugate gamma prior over $\mu_v$ directly and drew exact posterior samples to compute the information rate. We depict the results in Fig. 4(a).

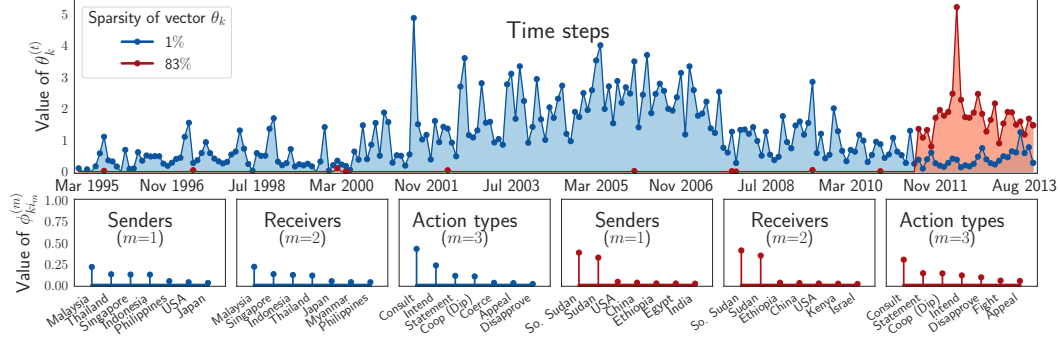

(a) We visualize two components inferred by a sparse variant of the PRGDS (i.e., $\epsilon_0^{(\theta)} = 0$) from the ICEWS data set of international events. The blue component was also inferred by the other models while the red component was not. The red component is specific to South Sudan, as revealed by visualizing the largest values of the sender and receiver factor vectors (bottom row, red). South Sudan was not a country until July 2011 when it gained independence from Sudan. The gamma states (top row, red) are therefore sparse—i.e., $\theta_k^{(t)} = 0$ in 94% of time steps (months) prior to July 2011 and in 83% of the time steps overall. In contrast, the blue component represents Southeast Asian relations, which are active in all time steps. The sparse variant can infer both temporally persistent latent structures (e.g., blue), as well as bursty latent structures that are highly localized in time (e.g., red).

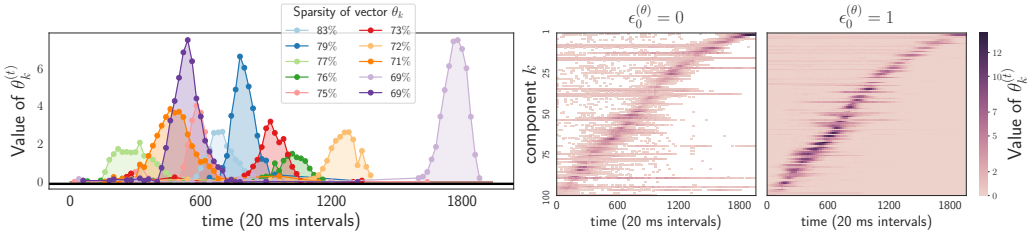

(b) We visualize components inferred by the PRGDS from the macaque motor cortex data set. The components inferred by a sparse variant (i.e., $\epsilon_0^{(\theta)} = 0$) are bursty and highly localized in time (left), suggesting that neurons may be tuned to specific periods of the trial. The $K \times T$ gamma latent states for this variant of the PRGDS are sparse (middle, white cells correspond to $\theta_k^{(t)} = 0$). The components (rows) are sorted by the time step in which the largest $\theta_k^{(t)}$ occurred, so the banded structure indicates that each component is only active for a short duration. In contrast, the components inferred by the non-sparse variant (i.e., $\epsilon_0^{(\theta)} = 1$) are active in all time steps (right).

Figure 5: The PRGDS is capable of inferring latent structures that are highly localized in time.

**Tensors.** We used two international event data sets—GDELT and ICEWS—where $y_{i \xrightarrow{a} j}^{(t)}$ is the number of times country $i$ took action $a$ toward country $j$ during time step $t$. Each data set consists of a sequence of count tensors, each of which contains the $V \times V \times A$ event counts for that time step, where $V = 249$ countries and $A = 20$ action types. For both data sets, we used months as time steps. For GDELT, we considered the date range 2003–2008, yielding $T = 72$; for ICEWS, we considered the date range 1995–2013, yielding $T = 228$. We also used a data set of multi-neuronal spike train recordings of macaque monkey motor cortexes [62, 63]. In this data set, a count $y_{ij}^{(t)}$ is the number of times neuron $i$ spiked in trial $j$ during time step $t$. These counts form a sequence of $N \times V$ matrices, where $N = 100$ is the number of neurons and $V = 1,716$ is the number of trials. We used 20-millisecond intervals as time steps, yielding $T = 162$. For each data set, we created three random masks, each corresponding to six heldout time steps in the range $[2, T-2]$. We fit each model to each data set and mask using two independent chains of 4,000 MCMC iterations, saving every $50^{\text{th}}$ posterior sample after the first 1,000 iterations to compute the information rate. We also fit BPTF using variational inference as described by Schein et al. [9], and then sampled from the fitted variational posterior to compute the information rate. Following Schein et al. [22], we set $K = 100$ for all models. We depict the results in Fig. 4(b), where the error bars reflect variability across the random masks.

**Quantitative results.** In all sixteen studies, the dynamic models outperform BPTF. In all but one study, the PGDS and a sparse variant of the PRGDS (i.e., $\epsilon_0^{(\theta)} = 0$) outperform the other models. For smoothing, the PRGDS performs better than or similarly to the PGDS. In five of the eight smoothing

studies, the sparse variant of the PRGDS obtains a higher information gain than the PGDS; in the remaining three smoothing studies, there is no discernible difference between the models. For forecasting, we find the converse relationship. In four of the eight forecasting studies, the PGDS obtains a higher information gain than the PGDS; in the remaining forecasting studies, there is no discernible difference. In all studies, the sparse variant of the PRGDS obtains better smoothing and forecasting performance than the non-sparse variant (i.e., $\epsilon_0^{(\theta)} = 1$). We conjecture that the better performance of the sparse variant can be explained by the form of the marginal expectation of $\boldsymbol{\theta}^{(t)}$ (see Eq. (8)). When $\epsilon_0^{(\theta)} > 0$ this expectation includes an additive term that grows as more time steps are forecast. When $\epsilon_0^{(\theta)} = 0$, this term disappears and the expectation matches that of the PGDS (see Eq. (10)).

**Qualitative analysis.** We also performed a qualitative comparison of the latent structures inferred by the different models and found that the sparse variant of the PRGDS inferred some components that the other models did not. Specifically, the sparse variant of the PRGDS is uniquely capable of inferring bursty latent structures that are highly localized in time; we visualize examples in Fig. 5. To compare the latent structures inferred by the PGDS and the PRGDS, we aligned the models' inferred components using the Hungarian bipartite matching algorithm [64] applied to the models' continuous gamma latent states. The $k^{\text{th}}$ component's activation vector $\boldsymbol{\theta}_k = (\theta_k^{(1)}, \ldots, \theta_k^{(T)})$ constitutes a signature of that component's activity; these signatures are sufficiently unique to facilitate alignment. In the supplementary material, we provide four components that are well aligned across the models. In Fig. 5(a), we visualize two components inferred by the sparse variant of the PRGDS; one of these components (blue) was also inferred by the other models, while the other component (red) was not.

## 6   Conclusion

We presented the Poisson-randomized gamma dynamical system (PRGDS), a tractable, expressive, and efficient model for sequentially observed count tensors. The PRGDS is based on a new modeling motif, an alternating chain of discrete Poisson and continuous gamma latent states that yields closed-form complete conditionals for all variables. We found that a sparse variant of the PRGDS, which allows the continuous gamma latent states to take values of exactly zero, often obtains better predictive performance than other models and infers latent structures that are highly localized in time.

**Acknowledgments**   We thank Saurabh Vyas, Alex Williams, and Krishna Shenoy for kindly providing us with the macaque monkey motor cortex data set and their corresponding preprocessing code. SWL was supported by the Simons Collaboration on the Global Brain (SCGB 418011). MZ was supported by NSF IIS-1812699. DMB was supported by ONR N00014-17-1-2131, ONR N00014-15-1-2209, NIH 1U01MH115727-01, NSF CCF-1740833, DARPA SD2 FA8750-18-C-0130, IBM, 2Sigma, Amazon, NVIDIA, and the Simons Foundation.

## Footnotes

[1] https://github.com/aschein/PRGDS

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
