[Supplementary Material · supplementary.pdf]

# Appendix for Poisson-Randomized Gamma Dynamical Systems

**Aaron Schein**
Data Science Institute
Columbia University

**Scott W. Linderman**
Department of Statistics
Stanford University

**Mingyuan Zhou**
McCombs School of Business
University of Texas at Austin

**David M. Blei**
Department of Statistics
Columbia University

**Hanna Wallach**
Microsoft Research
New York, NY

## 1 Shifted confluent hypergeometric (SCH) distribution

The SCH distribution arises in the context of Poisson–gamma–Poisson chains. Consider the following generative process for count $m$ involving latent variables $\theta$ and $h$ and fixed $c_1, c_2, c_3, \epsilon_0^{(\theta)} > 0$ :

$$m \sim \text{Pois}\left(\theta c_3\right), \tag{1}$$

$$\theta \sim \text{Gam}\left(\epsilon_0^{(\theta)} + h, c_2\right), \tag{2}$$

$$h \sim \text{Pois}\left(c_1\right). \tag{3}$$

As stated in the main paper, when $\epsilon_0 = 0$, a Gibbs sampler based on sampling $h$ and $\theta$ from their complete conditionals violates detailed balance since $h \stackrel{\text{a.s.}}{=} 0$ if $\theta = 0$, and vice versa. Instead, we should sample $h$ from its *incomplete* conditional—i.e., its distribution conditioned on all variables in its Markov blanket except $\theta$:

$$P(h \mid \epsilon_0^{(\theta)} = 0, -\backslash \theta) \triangleq \int P(h, \theta \mid \epsilon_0^{(\theta)} = 0, -)\, \mathbf{d}\theta. \tag{4}$$

Integrating $\theta$ out of the generative process given in Equations (1) to (3) yields the following generative process for $m$ as a negative binomial random variable, where $p \triangleq \frac{c_3}{c_3 + c_2}$:

$$m \sim \text{NB}\left(h,\, p\right), \tag{5}$$

$$h \sim \text{Pois}\left(c_1\right). \tag{6}$$

By Bayes' rule, the posterior of $h$ given $m$ is equal to:

$$P(h \mid m, c_1, p) = \frac{\text{Pois}\left(h; c_1\right) \text{NB}\left(m; h, p\right)}{P(m \mid c_1, p)}. \tag{7}$$

To find a closed form for this expression we need a closed form for the denominator. When the negative binomial has a count-valued first parameter, it is referred to as the *Pascal distribution* [1]. The construction in Equations (5) to (6) describes a Pascal variable with a Poisson-distributed first parameter—the marginal distribution of $m$ with $h$ marginalized out has been called the *Poisson–Pascal distribution* [2], which is a special case of the *Polya–Aeppli distribution* [1]:

$$P(m \mid c_1, p) = \sum_{h=0}^{\infty} \text{Pois}\left(h; c_1\right) \text{NB}\left(m; h, p\right) \tag{8}$$

$$= \text{Polya-Aeppli}\left(m;\, c_1, p\right). \tag{9}$$

The Polya-Aeppli distribution is defined by two parameters—$p \in (0,1)$ and $c \geq 0$—and PMF:

$$\text{Polya-Aeppli}\,(m;\, c, p) = \begin{cases} e^{-p\,c} & \text{if } m=0 \\ e^{-c_1} c\, p^m (1-p)\, {}_1\text{F}_1\big(m+1; 2; c(1-p)\big) & \text{otherwise,} \end{cases} \tag{10}$$

where ${}_1\text{F}_1\big(a; b; z\big)$ is Kummer's confluent hypergeometric function [3].

Plugging in the Polya-Aeppli PMF into the denominator of Eq. (7) (and the Poisson and negative binomial PMFs into the numerator) we obtain a closed-form expression for the posterior of $h$. Since the Polya-Aeppli's PMF is different for $m=0$ and $m>0$, we first consider the case where $m=0$:

$$P(h \,|\, m=0, c_1, p) = \frac{\text{Pois}\,(h; c_1)\, \text{NB}\,(0; h, p)}{\text{Polya-Aeppli}(0; c_1, p)} \tag{11}$$

$$= \frac{\frac{(c_1)^h}{h!} e^{-c_1} (1-p)^h}{e^{-p\,c_1}} \tag{12}$$

$$= \frac{[c_1(1-p)]^h}{h!} e^{-c_1(1-p)}. \tag{13}$$

We recognize this as the form of a Poisson PMF with parameter $\zeta \triangleq c_1(1-p)$:

$$= \text{Pois}\,(h; \zeta). \tag{14}$$

Thus, when $m=0$, the posterior of $h$ is Poisson. The posterior of $h$ when $m>0$ is:

$$P(h \,|\, m>0, c_1, p) = \frac{\text{Pois}\,(h; c_1)\, \text{NB}\,(m; h, p)}{\text{Polya-Aeppli}(m; c_1, p)} \tag{15}$$

$$= \frac{\frac{(c_1)^h}{h!} e^{-c_1} \frac{\Gamma(m+h)}{m!\Gamma(h)} p^m (1-p)^h}{e^{-c_1} c_1\, p^m (1-p)\, {}_1\text{F}_1\big(m+1; 2; c_1(1-p)\big)} \tag{16}$$

$$= \frac{\frac{\Gamma(m+h)}{h!\, m!\Gamma(h)} [c_1(1-p)]^{h-1}}{{}_1\text{F}_1\big(m+1; 2; c_1(1-p)\big)}. \tag{17}$$

Since $c_1$ and $h$ always appear together, we plug in $\zeta$ as defined in Eq. (14), to obtain

$$= \frac{\frac{\Gamma(m+h)}{h!\, m!\Gamma(h)} \zeta^{h-1}}{{}_1\text{F}_1\big(m+1; 2; \zeta\big)}, \tag{18}$$

which is a discrete distribution defined by two parameters—$\zeta > 0$ and $m \in \{1, 2, \dots\}$. When $m > 0$, $h \overset{\text{a.s.}}{>} 0$ since $m \overset{\text{a.s.}}{=} 0$ if $h=0$. Thus, this distribution is defined on the support $h \in \{1, 2, \dots\}$. What is this distribution? It is illustrative to consider its probability generating function (PGF):

$$G(s) = \mathbb{E}\big[s^h \,|\, m, \zeta\big] \tag{19}$$

$$= \sum_{h=1}^{\infty} s^h \frac{\frac{\Gamma(m+h)}{h!\, m!\Gamma(h)} \zeta^{h-1}}{{}_1\text{F}_1\big(m+1; 2; \zeta\big)} \tag{20}$$

$$= s\, \frac{{}_1\text{F}_1\big(m+1; 2; s\zeta\big)}{{}_1\text{F}_1\big(m+1; 2; \zeta\big)}. \tag{21}$$

The PGF in Eq. (21) nearly matches that of the *confluent hypergeometric distribution* [1]. The confluent hypergeometric distribution $h \sim \text{ConfHyp}(h; a, b, z)$ is a discrete distribution over counts $h \in \{0, 1, 2, \dots\}$ defined by three parameters $a, b, z > 0$ and PGF equal to $G'(s) = \frac{{}_1\text{F}_1(a;b;sz)}{{}_1\text{F}_1(a;b;z)}$. The $s$ out in front of the PGF in Eq. (21) is the only difference between it and the PGF of a confluent hypergeometric distribution with parameters $a=m+1$, $b=2$, and $z=\zeta$. However, the following

manipulation reveals that the PGF in Eq. (21) defines a *shifted* confluent hypergeometric distribution:

$$G(s) = sG'(s) \tag{22}$$

$$= s\sum_{h=0}^{\infty} s^h \, \mathrm{ConfHyp}(h; m+1, 2, \zeta) \tag{23}$$

$$= \sum_{h=1}^{\infty} s^h \, \mathrm{ConfHyp}(h-1; m+1, 2, \zeta). \tag{24}$$

The posterior distribution of $h$ when $m > 0$ can thus appropriately be described as a *shifted confluent hypergeometric (SCH) distribution*. An SCH random variable $h \sim \mathrm{SCH}(m, \zeta)$ can be generated as $h \triangleq n+1$ where $n \sim \mathrm{ConfHyp}(m+1, 2, \zeta)$.

## 1.1 Proof of Theorem 1

**Theorem 1:** *The incomplete conditional* $P(h \,|\, \epsilon_0^{(\theta)} = 0, -\backslash\theta) \triangleq \int P(h, \theta \,|\, \epsilon_0^{(\theta)} = 0, -) \, \mathrm{d}\theta$ *is*

$$(h \,|\, -\backslash\theta) \sim \begin{cases} \mathrm{Pois}\left(\frac{c_1 c_2}{c_3 + c_2}\right) & \text{if } m = 0 \\ \mathrm{SCH}\left(m, \frac{c_1 c_2}{c_3 + c_2}\right) & \text{otherwise.} \end{cases} \tag{25}$$

**Proof:** The preceding derivation constitutes the proof—in particular, see Eq. (14) and Eq. (18).

## 1.2 Sampling from the SCH distribution

As stated above, an SCH random variable can be generated in terms of a confluent hypergeometric random variable. However, we are unaware of any open-source implementation for sampling from the confluent hypergeometric distribution.

We implement a table sampler for the SCH distribution by directly evaluating its PMF at candidate values. This sampler is efficient if we begin with mode $h^*$ as the first candidate value and then step out $h^* - 1$ or $h^* + 1$ (if the mode is not accepted). Since the confluent hypergeometric distribution is unimodal and underdispersed [1], the SCH is as well—thus, a table sampler that begins at the mode frequently terminates after a small number iterations, since the PMF quickly and monotonically decays in both directions from the mode.

To derive the mode of the SCH, we appeal to the fact that any PMF has the following property,

$$P(H = h^* - 1) \leq P(H = h^*) \geq P(H = h^* + 1), \tag{26}$$

which can be equivalently stated in terms of the following two equations:

$$\frac{P(H = h^*)}{P(H = h^* - 1)} \geq 1, \tag{27}$$

$$\frac{P(H = h^*)}{P(H = h^* + 1)} \leq 1. \tag{28}$$

Plugging in the PMF of the SCH distribution we obtain the following two inequalities:

$$\frac{\zeta(h^* + m - 1)}{h^*(h^* - 1)} \geq 1, \tag{29}$$

$$\frac{\zeta(h^* + m)}{h^*(h^* + 1)} \leq 1. \tag{30}$$

Solving this system of inequalities gives us the following bounds on $h^*$:

$$f(\zeta, m) - 0.5 \leq h^* \leq f(\zeta, m) + 0.5, \tag{31}$$

where $f(\zeta, m) \triangleq \frac{1}{2}\left(\sqrt{2\zeta(2m-1) + \zeta^2 + 1} + \zeta\right)$. Since $h$ discrete, the mode of the SCH is

$$\mathrm{mode}\,(h; m, \zeta) = \left\lfloor \frac{1}{2}\left(\sqrt{2\zeta(2m-1) + \zeta^2 + 1} + \zeta\right) \right\rfloor, \tag{32}$$

which does involve any special functions and is thus efficient to compute.

## 2 Closed-form complete conditionals for the PRGDS

Recall that the per-component weights $\lambda_k$ appear in the Poisson rate of each observed count $y_{\mathbf{i}}^{(t)} \sim \text{Pois}\left(\rho^{(t)} \sum_{k=1}^{K} \lambda_k \theta_k^{(t)} \prod_{m=1}^{M} \phi_{ki_m}^{(m)}\right)$ as well as in the Poisson rate of the first latent discrete state $h_{k\cdot}^{(1)} \sim \text{Pois}\left(\tau \sum_{k_2=1}^{K} \pi_{kk_2} \lambda_{k_2}\right)$. Consider the following sum of latent sources $y_{\cdot k}^{(\cdot)} \triangleq \sum_{t=1}^{T} \sum_{\mathbf{i}} y_{\mathbf{i}k}^{(t)}$—it is a Poisson random variable $y_{\cdot k}^{(\cdot)} \sim \text{Pois}\left(\lambda_k \omega_k\right)$ where $\omega_k \triangleq \prod_{m=1}^{M} \phi_{k\cdot}^{(m)} \sum_{t=1}^{T} \rho^{(t)} \theta_k^{(t)}$. Now define $h_{\cdot k}^{(1)} \triangleq \sum_{k_1=1}^{K} h_{k_1 k}^{(1)}$ to be the sum of the $k^{\text{th}}$ column of the first ($t=1$) matrix of latent counts—it is distributed $h_{\cdot k}^{(1)} \sim \text{Pois}\left(\lambda_k \tau \pi_{\cdot k}\right)$. Finally, define the sum $m_k^{(\lambda)} \triangleq h_{\cdot k}^{(1)} + y_{\cdot k}^{(\cdot)}$ which isolates all dependence on $\lambda_k$ and is Poisson $m_k^{(\lambda)} \sim \text{Pois}\left(\lambda_k(\tau \pi_{\cdot k} + \omega_k)\right)$. By gamma–Poisson conjugacy, the complete conditional for $\lambda_k$ is thus

$$\left(\lambda_k \mid -\right) \sim \text{Gam}\left(\epsilon_0^{(\lambda)} + g_k + m_k^{(\lambda)}, \beta + \tau \pi_{\cdot k} + \omega_k\right), \tag{33}$$

$$m_k^{(\lambda)} \triangleq \left(\sum_{t=1}^{T} \sum_{\mathbf{i}} y_{\mathbf{i}k}^{(t)}\right) + \left(\sum_{k_1=1}^{K} h_{k_1 k}^{(1)}\right), \tag{34}$$

$$\omega_k \triangleq \prod_{m=1}^{M} \phi_{k\cdot}^{(m)} \sum_{t=1}^{T} \rho^{(t)} \theta_k^{(t)}. \tag{35}$$

We may apply the identifies on Poisson–gamma–Poisson chains provided in the main paper to derive the complete conditional for $g_k$ when $\epsilon_0^{(\lambda)} > 0$ as

$$\left(g_k \mid -\right) \sim \text{Bessel}\left(\epsilon_0^{(\lambda)} - 1, 2\sqrt{\lambda_k \beta \frac{\gamma}{K}}\right), \tag{36}$$

and for $\epsilon_0^{(\lambda)} = 0$ as

$$\left(g_k \mid -\right) \sim \text{SCH} \begin{cases} \text{Pois}\left(\zeta_k^{(\lambda)}\right) & \text{if } m_k^{(\lambda)} = 0 \\ \text{SCH}\left(m_k^{(\lambda)}, \zeta_k^{(\lambda)}\right) & \text{otherwise,} \end{cases} \tag{37}$$

$$\zeta_k^{(\lambda)} \triangleq \frac{\beta \frac{\gamma}{K}}{\tau \pi_{\cdot k} + \omega_k + \beta}. \tag{38}$$

By gamma–Poisson and gamma–gamma conjugacy the complete conditionals for $\gamma$ and $\beta$ are

$$\left(\gamma \mid -\right) \sim \text{Gam}\left(a_0 + g_\cdot, b_0 + 1\right), \tag{39}$$

$$\left(\beta \mid -\right) \sim \text{Gam}\left(\alpha_0 + K\epsilon_0^{(\lambda)} + g_\cdot, \alpha_0 + \lambda_\cdot\right). \tag{40}$$

By both gamma–gamma and gamma–Poisson conjugacy, the complete conditional for $\tau$ is gamma:

$$\left(\tau \mid -\right) \sim \text{Gam}\left(\alpha_0 + TK\epsilon_0^{(\theta)} + 2\,h_{\cdot\cdot}^{(\cdot)}, \alpha_0 + \lambda_\cdot + \theta_\cdot^{(\cdot)} + \sum_{k=1}^{K} \sum_{t=2}^{T-1} \sum_{k_2=1}^{K} \pi_{kk_2} \theta_{k_2}^{(t-1)}\right). \tag{41}$$

By Dirichlet–multinomial conjugacy, the complete conditional for $\boldsymbol{\pi}_k$ is Dirichlet:

$$\left(\boldsymbol{\pi}_k \mid -\right) \sim \text{Dir}\left(a_0 + h_{1k}^{(\cdot)}, \ldots, a_0 + h_{Kk}^{(\cdot)}\right). \tag{42}$$

By Dirichlet–multinomial conjugacy, the complete conditional for each factor vector $\boldsymbol{\phi}_k^{(m)}$ is

$$\left(\boldsymbol{\phi}_k^{(m)} \mid -\right) \sim \text{Dir}\left(a_0 + \sum_{\mathbf{i}:\mathbf{i}_m=1} y_{\mathbf{i}k}^{(t)}, \cdots, a_0 + \sum_{\mathbf{i}:\mathbf{i}_m=L_m} y_{\mathbf{i}k}^{(t)}\right), \tag{43}$$

where the sum $\sum_{\mathbf{i}:\mathbf{i}_m=d}$ sums over all values of the multi-index $\mathbf{i} = (\mathbf{i}_1, \ldots, \mathbf{i}_M)$ that have the $m^{\text{th}}$ index equal to a specific value $\mathbf{i}_m = d$.

By gamma–Poisson conjugacy, the complete conditional for $\rho^{(t)}$ or $\rho$ (for the stationary variant) are

$$\left(\rho^{(t)} \mid -\right) \sim \text{Gam}\left(a_0 + y_\cdot^{(t)}, b_0 + \omega^{(t)}\right), \tag{44}$$

$$\left(\rho \mid -\right) \sim \text{Gam}\left(a_0 + y_\cdot^{(\cdot)}, b_0 + \sum_{t=1}^{T} \omega^{(t)}\right), \tag{45}$$

where $\omega^{(t)} \triangleq \sum_{k=1}^{K} \lambda_k \prod_{m=1}^{M} \phi_{k\cdot}^{(m)} \theta_k^{(\cdot)}$.

# 3 Tensor generalization of the PGDS

## 3.1 Original generative process

Schein et al. (2016) [4] originally introduced the PGDS to model $T \times V$ count matrices $Y$—the PGDS assumes each count $y_v^{(t)}$ in the matrix is a Poisson random variable:

$$y_v^{(t)} \sim \text{Pois}\Big(\rho^{(t)} \sum_{k=1}^{K} \theta_k^{(t)} \phi_{kv}\Big). \tag{46}$$

The states $\theta_k^{(t)}$ evolve as

$$\theta_k^{(t)} \sim \text{Gam}\left(\tau \sum_{k_2=1}^{K} \pi_{kk_2} \theta_{k_2}^{(t-1)}, \tau\right). \tag{47}$$

The columns of the factor matrix $\Phi$ are Dirichlet distributed:

$$\phi_k \sim \text{Dir}\left(a_0, \ldots a_0\right). \tag{48}$$

See the original paper [4] for more details.

## 3.2 Generative process for tensor generalization

The PGDS can be generalized to be a canonical polyadic (CP) decomposition [5] of sequentially observed $M$-mode tensors by assuming each count $y_{\mathbf{i}}^{(t)}$ is

$$y_{\mathbf{i}}^{(t)} \sim \text{Pois}\Big(\rho^{(t)} \sum_{k=1}^{K} \theta_k^{(t)} \prod_{m=1}^{M} \phi_{k\mathbf{i}_m}^{(m)}\Big). \tag{49}$$

The states $\theta_k^{(t)}$ evolve the same as in Eq. (47). There are now $M$ different factor matrices—the columns of the $m^{\text{th}}$ matrix $\Phi^{(m)}$ are Dirichlet distributed:

$$\phi_k^{(m)} \sim \text{Dir}\left(a_0, \ldots a_0\right). \tag{50}$$

All other aspects are the same as the matrix version.

## 3.3 Complete conditionals

The latent sources for the tensor PGDS have the following complete conditional:

$$\left((y_{\mathbf{i}k}^{(t)})_{k=1}^{K} \mid -\right) \sim \text{Multinom}\left(y_{\mathbf{i}}^{(t)}, \left(\theta_k^{(t)} \prod_{m=1}^{M} \phi_{k\mathbf{i}_m}^{(m)}\right)_{k=1}^{K}\right). \tag{51}$$

By Dirichlet–multinomial conjugacy, each column of the $m^{\text{th}}$ has the following complete conditional:

$$\left(\phi_k^{(m)} \mid -\right) \sim \text{Dir}\left(a_0 + \sum_{\mathbf{i}:\mathbf{i}_m=1} y_{\mathbf{i}k}^{(t)}, \cdots, a_0 + \sum_{\mathbf{i}:\mathbf{i}_m=L_m} y_{\mathbf{i}k}^{(t)}\right), \tag{52}$$

where the sum $\sum_{\mathbf{i}:\mathbf{i}_m=d}$ sums over all values of the multi-index $\mathbf{i} = (\mathbf{i}_1, \ldots, \mathbf{i}_M)$ that have the $m^{\text{th}}$ index equal to a specific value $\mathbf{i}_m = d$.

All other complete conditionals are the same as in matrix version (see the original paper).

# 4 Qualitative analysis of latent structure inferred from ICEWS data

We qualitatively compared the latent structure inferred by the two PRGDS variants and the PGDS on ICEWS international events data. To do so, we aligned the inferred components of one model to another using the Hungarian bipartite matching algorithm [6] applied to their inferred $K \times T$ gamma state matrices. The $k^{\text{th}}$ component's activation vector $\boldsymbol{\theta}_k = (\theta_k^{(1)}, \dots, \theta_k^{(T)})$ constitutes a signature of that component's activity; these signatures are sufficiently unique to facilitate alignment.

We interpret the components as *multilateral relations* [7], where a component is characterized by its activation vector $\boldsymbol{\theta}_k$ (i.e., when that component is active), who the typical sender $\phi_k^{(1)}$ and receiver $\phi_k^{(2)}$ countries are, and what action types $\phi_k^{(3)}$ are typically used. We found the vast majority of inferred components to be well aligned across all three models. In Figures 1 to 4, we provide four examples of components inferred by each of the three models that were aligned to each other by the matching algorithm. We visualize each component's $\boldsymbol{\theta}_k$ in chronological order in the top panel of each plot. The bottom-left stem plot displays the top values of sender parameters, in descending order. If fewer than ten senders account for more than 99% of the mass, we only display their names; otherwise, the top seven are given. The same is true for the bottom-middle and bottom-right stem plots, corresponding to receivers and action types. We see that all four aligned components measure a qualitatively similar multilateral relation corresponding respectively to the Israeli–Palestinian conflict (Fig. 1), Vietnamese international relations (Fig. 2), Central European relations (Fig. 3), and West African relations (Fig. 4).

There were only a few instances where the aligned components were qualitatively dissimilar. In particular, we found a few cases where the aligned components of the PGDS and the non-sparse variant of the PRGDS were qualitatively similar, but the component inferred by the sparse variant of the PRGDS had no counterpart. This occurred when the component inferred by the sparse variant featured a highly localized pattern. The component visualized in the main text is such an example.

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

(a) Component inferred by the sparse PRGDS ($\epsilon_0^{(\theta)} = 0$).

(b) Component inferred by the non-sparse PRGDS ($\epsilon_0^{(\theta)} = 1$).

(c) Component inferred by the PGDS.

Figure 1: A component aligned across all three models that measures the Israeli–Palestinian conflict.

(a) Component inferred by the sparse PRGDS ($\epsilon_0^{(\theta)} = 0$).

(b) Component inferred by the non-sparse PRGDS ($\epsilon_0^{(\theta)} = 1$).

(c) Component inferred by the PGDS.

Figure 2: A component aligned across all three models that measures Vietnamese relations.

(a) Component inferred by the sparse PRGDS ($\epsilon_0^{(\theta)} = 0$).

(b) Component inferred by the non-sparse PRGDS ($\epsilon_0^{(\theta)} = 1$).

(c) Component inferred by the PGDS.

Figure 3: A component aligned across all three models that measures Central European relations.

(a) Component inferred by the sparse PRGDS ($\epsilon_0^{(\theta)} = 0$).

(b) Component inferred by the non-sparse PRGDS ($\epsilon_0^{(\theta)} = 1$).

(c) Component inferred by the PGDS.

Figure 4: A component aligned across all three models that measures West African relations.