[Reviews · NeurIPS 2019]

Reviewer 1



This is a nicely written paper. The additions lead to a sampler which is simpler to implement. The method is shown to perform much better than the PGDS model in some data sets. Here are a few comments: 1) the transition for \theta_t (summing over h_k^{(t)}) is mixture of gamma distribution (in contrast to the PGDS model which assumes a single gamma distribution). It would be interesting to see what effects this has on the properties of the model. There is talk of "bursty" behaviour but I didn't see this effect being clearly explored in the paper. 2) Why is \epsilon_0^{(g)}=1 used as a canonical choice when this parameter is non-zero. Why? Would it make sense to put a prior on this parameter and learn it from the data. 3) The MCMC sampler is simpler to implement but I didn't see any discussion of the timings of the different methods. It would be interesting to know if there were differences in the computational times. 4) The sparse PrGDS performs a lot better than the non-sparse version on the GDELT, ICEWS (as matrices), NeurIPS and DBLP data sets but the performance is similar for the other data sets. It would be interesting to know what aspect of the data favour the sparse version (other, presumably, the DGP being "sparse"). Why is there a difference in performance for the ICEWS data when looking matrices but not tensor. 5) The PrGDS outperforms the PGDS for the GDELT and ICEWS (as matrices) but gives similar performance on other data sets. Is there some way to know from the data when the PrGDS will perform much better than PGDS? Originality: This is an extension of the PGDS model but I think that this is an important development since it allows the introduction of sparsity which leads to much better predictive performance for some problems. Quality: A lot of work has obviously gone in to this paper. The construction of the Gibbs sampler is neat, the method is well-explained and there are a range of comparisons. My one main criticism is that the competing benefits of the two methods could be better explored. Clarity: The paper is well-written and clear. Significance: I think that this is a nice complementary idea to the PGDS which add some extra sparsity into the mix. This clearly leads to better predictive performance in some data sets.

Reviewer 2



The contribution builds on the recent work on Poisson Gamma Dynamical Systems by Schein et al. and its deep variants albeit departing from the standard PGDS formulation and required augmentation scheme by introducing alternating chains of Poisson and Gamma latent state variables. This expanded construction offers closed-form conditionals via the relatively unknown Bessel distribution and the authors defined SCH distribution whose MGF is derived. The paper offers significant novelty and extends the state of the art in the area. A nice characteristic of the contribution is that although the latent structure is expanded (in relation to PGDS) in the model (and hence its complexity and any identifiability issues), the induced sparsity in the latent states and the closed form conditionals from the Poisson-gamma chaining simplifies inference and helps (?) with identifiability. The paper is well written, the experiments are convincing and demonstrate improvement over the PGDS and a simple dynamic baseline (GP-DPFA). I appreciate the accompanying code for both PGDS and PrGDS and the expanded material in the appendix. My only minor concerns are: - Identifiability: There is little mention (if any) of this in this line of work and more complex constructions of latent state chaining will suffer further on this aspect. Is the model identifiable and to what extend? Does the induced sparsity help? Is that shown anywhere?And how does that relate to the observations made in Sec. 4 of the appendix? - Computational complexity: please provide this in comparison to PGDS - MCMC: Any evidence of convergence of the chain?any rates? - MCMC: Since you have a Gibbs sampler you should be able to also derive an SVI or EM variant? - hyper-parameters: why is \epsilon_0^{(\lambda)} kept at 1 and would a 0 induce further sparsity on the component-weights? - "perplexity": This is mentioned from the start but only defined in experiments section. A pointer to that would help in the start. More importantly: Is this the right measure to be using for a probabilistic model like this? wouldn't you be interested in more Bayesian measures like the coverage you are offering? See for example: Leininger, T. J., Gelfand, A. E., et al. (2017). Bayesian inference and model assessment for spatial point pat- terns using posterior predictive samples. Bayesian Analysis, 12(1):1–30. **Post author response** The authors have clarified and responded to my questions for MCMC convergence and time complexity, and also supported their use of perplexity. It is not clear if alternative metrics that focus on UQ like coverage will be offered in the manuscript or if any identifiability discussion will be offered though. I retain my accept score of 7 and encourage the authors to consider in the final submission the 2 topics above.

Reviewer 3



I've read the authors' response and my score remain the same. The mistake in the comments is corrected, thanks for the remind. ----------------------------------------------------------------------------------------------- This paper uses a new trick on the Poisson Gamma Dynamic Systems to achieve tractable and more expressive models. Experimental results verify the advantages of the newly proposed model and look to be promising. The poisson-gamma-poisson motif proposed in this paper contains substantial originality. In my rough understanding, this technique can be readily applied to other models (e.g. Gamma Belief Networks, maybe Dirichlet Belief Networks) and circumvent the complex data augmentation techniques usually required. Thus, this paper will have impact on the community. Since this trick introduces an "auxiliary" discrete variable to the standard PGDS, I am interested in the sampling efficiency with comparison to the standard PGDS. I think should be ok, just to confirm. I like this paper, especially the poisson-gamma-poisson trick. Maybe the only pity is that this trick only applies on the rate parameter of Gamma distribution.

[Author Response · NeurIPS 2019]

Thank you to all the reviewers for their careful evaluation and thoughtful feedback. We were happy to see that all three reviewers expressed appreciation for the paper's clarity and theoretical novelty and believed our experimental results were strong and corroborated the theoretical claims. R1 believes the model is "an important development" while R2 says the paper "offers significant novelty and extends the state of the art in the area" and R3 says it "contains substantial novelty" and "will have an impact on the community". We found all three reviewers understood and appreciated the main arguments and technical details of the paper. We again thank all three reviewers for their careful reviewing work.

**Time complexity.** All three reviewers asked about the time complexity of our model versus PGDS. The two have the same time complexity. We will update the paper to emphasize this point. We state it in section 3 (line 143) where we say that any Poisson factorization model which yields the multinomial in equation 11 scales linearly with the non-zero counts—i.e., $\mathcal{O}(S\,K)$ where $S$ is the number of non-zeros and $K$ is the number of components. PrGDS and PGDS have the same complexity but different constants—the difference is that MCMC for PGDS involves sampling $T \times K$ "auxiliary" counts from the Chinese restaurant table (CRT) distribution while PrGDS involves sampling $T \times K$ counts from the Bessel or SCH distribution. The CRT, Bessel, and SCH can all be sampled in constant time with similar constants since they are all underdispersed unimodal distributions whose PMFs and modes can are available in closed form.

**Relationship between PrGDS and PGDS.** We will update the paper to clarify the relationship between PGDS and PrGDS since it seems that both R1 and R3 have a subtle misunderstanding of it that may have led them to down-weight the originality of the paper. R1 says "Originality: This is an extension of the PGDS model." R3 says: "This paper uses a new trick on [PGDS]". PrGDS is closely related to PGDS but it is neither an "extension" nor a "trick" for it. We would like to highlight R2's characterization: "[PrGDS] builds on [PGDS] albeit departing from the standard PGDS formulation and required augmentation scheme..." The key point is that the proposed model does not have "auxiliary" variables but rather the "the latent structure is expanded (in relation to PGDS)", as R2 correctly states. The proposed model's expanded latent structure includes an extra layer of latent states and thus expresses more dispersed processes. This may answer R1's related question: "the transition for $\theta_k^{(t)}$ is a mixture of gammas in contrast to PGDS...it would be interesting to see what effects this has...". We agree and characterize this mixture in equation 9 and figure 2—it can be understood as an overdispersed gamma. Our model, by extension, can be understood as an overdispersed PGDS.

**Hyperparameters $\epsilon_0^{(\lambda)}$ and $\epsilon_0^{(\theta)}$.** R1 and R2 ask about hyperparameters. For theoretical reasons given in the Discussion, we believe that $\epsilon_0^{(\theta)}=0$ should perform the best. We thus selected a simple alternative $\epsilon_0^{(\theta)}=1$ as an illustrative baseline to corroborate the theory. There is no conjugate prior, but we agree that inferring it would be interesting and are currently working on an auxiliary variable scheme to do so. We also fixed $\epsilon_0^{(\lambda)}=1$ to limit the number of branching paths for the purpose of clean exposition but agree that sparsity in the component weights is another interesting avenue.

**R1.** R1 asks about "burstiness". We use the term, like Schein et al. (2016) and others, to refer to non-smooth time series that may exhibit extreme values that are immediately preceded by small or zero values. R1 asks about the difference in performance across different data sets. Aggregating data into matrices versus tensors yields count sequences of differing levels of "burstiness" and sparsity, which we believe to be the contributing factors to differences in performance. We agree with R1 that it would be interesting to precisely characterize when the performance of PrGDS and PGDS will differ.

**R2.** R2 asks about perplexity. This metric is commonly used within the topic modeling community—but, we will make it clearer that it has a one-to-one relationship with posterior predictive probability, which is standard throughout Bayesian machine learning. For a heldout count $y_i$ and training data $Y$ the posterior predictive $P(y_i \,|\, Y) \approx \frac{1}{S}\sum_{s=1}^{S} P(y_i \,|\, \mu_s)$ can be approximated with $S$ samples $\mu_s \sim P(\mu \,|\, Y)$ drawn from the posterior. Line 212 in the paper then shows how perplexity is inversely proportional to the posterior predictive. We agree that coverage would be another illuminating metric. R2 also makes a very intriguing suggestion about whether the sparsity of PrGDS may assist in showing identifiability; we don't have such results now, but will think about it for future work. R2 also asks about MCMC. In figure 1, we provide some evidence of convergence—we found that all models converged on all data sets before 1,000 iterations, which is why we discarded the first 1,000 samples as burn-in in the experiments. R2 also mentions variational inference—yes, we have derived VI updates from the Gibbs sampler and are currently working on a follow-up paper!

Figure 1: Four chains of the sparse PrGDS on ICEWS tensor data—all converge after 750 iterations.

**R3.** R3 makes an excellent point: "this technique can be readily applied to other models (e.g., Gamma belief networks, maybe Dirichlet Belief Networks) and circumvent the complex data augmentation techniques usually required." Indeed, the reason we chose to highlight the Poisson-gamma-Poisson motif in its own section 4.1 is because of its potential application to a wide variety of new models. However, R3 also says "a 7 is the maximum I can give...as this trick applies only to the rate parameter of the Gamma". Our construction applies to the *shape*, not rate, which is the more challenging (non-conjugate) parameter to infer. We hope R3 will not limit their score to a 7 given that elegant and efficient solutions to gamma shape inference have many possible applications (some of which R3 themself suggests!).

[Meta-Review · NeurIPS 2019]

This is a solid contribution that proposes a novel model for sequential count tensor data, extending the existing Poisson-gamma dynamical system. The new model accounts for sparsity, and has tractable conditional distributions that enable the implementation of a Gibbs sampler. The experiments are convincing and demonstrate improvement over the PGDS and other baseline.